# PFKFB3 Inhibition Impairs Erlotinib-Induced Autophagy in NSCLCs

**DOI:** 10.3390/cells10071679

**Published:** 2021-07-03

**Authors:** Nadiia Lypova, Susan M. Dougherty, Lilibeth Lanceta, Jason Chesney, Yoannis Imbert-Fernandez

**Affiliations:** 1Department of Medicine, School of Medicine, University of Louisville, Louisville, KY 40202, USA; nadiia.lypova@louisville.edu (N.L.); susan.dougherty@louisville.edu (S.M.D.); lilibeth.lanceta@louisville.edu (L.L.); 2James Graham Brown Cancer Center, School of Medicine, University of Louisville, Louisville, KY 40202, USA

**Keywords:** autophagy flux, lung cancer, epidermal growth factor receptor (EGFR), 6-phosphofructo-2-kinase/fructose-2,6-bisphosphatase 3 (PFKFB3), cytoprotective autophagy

## Abstract

Tyrosine kinase inhibitors (TKIs) targeting the kinase domain of the epidermal growth factor receptor (EGFR), such as erlotinib, have dramatically improved clinical outcomes of patients with EGFR-driven non-small cell lung carcinomas (NSCLCs). However, intrinsic or acquired resistance remains a clinical barrier to the success of FDA-approved EGFR TKIs. Multiple mechanisms of resistance have been identified, including the activation of prosurvival autophagy. We have previously shown that the expression and activity of PFKFB3—a known driver of glycolysis—is associated with resistance to erlotinib and that PFKFB3 inhibition improves the response of NSCLC cells to erlotinib. This study focuses on investigating the role of PFKFB3 in regulating erlotinib-driven autophagy to escape resistance to erlotinib. We evaluated the consequence of pharmacological inhibition of PFKFB3 on erlotinib-driven autophagy in NSCLC cells with different mutation statuses. Here, we identify PFKFB3 as a mediator of erlotinib-induced autophagy in NSCLCs. We demonstrate that PFKFB3 inhibition sensitizes NCSLCs to erlotinib via impairing autophagy flux. In summary, our studies uncovered a novel crosstalk between PFKFB3 and EGFR that regulates erlotinib-induced autophagy, thus contributing to erlotinib sensitivity in NSCLCs.

## 1. Introduction

Activating mutations in the kinase domain of the epidermal growth factor receptor (EGFR) drive the growth of 10–30% of non-small cell lung cancers (NSCLCs), the leading cause of cancer mortality worldwide with a five-year survival rate of 23% [1]. Targeted therapies, such as EGFR tyrosine kinase inhibitors (TKIs), have been successfully implemented for the treatment of NSCLC in patients with advanced or metastatic NSCLC carrying EGFR mutations. However, three generations of FDA-approved EGFR TKIs prove ineffective within 8 to 16 months due to intrinsic or acquired resistance [2,3]. Multiple biological processes, including bypass signaling and autophagy induction, limit the clinical efficacy of TKIs and decrease patient survival rates [4,5,6,7]. Therefore, elucidating the underlying mechanisms promoting therapy resistance represents a key step towards improving TKI therapeutic efficacy and overcoming drug resistance.

Accumulating evidence indicates that induction of macroautophagy (hereafter referred as autophagy) is a key resistance mechanism in multiple cancer types. NSCLC patients with autophagy-high tumors undergoing tumor resection followed by adjuvant chemotherapy had significantly lower 5-year survival rate [8]. Furthermore, low PARP1 expression and high p62 expression were associated with good outcomes among patients with NSCLC after EGFR TKI therapy [9]. Autophagy is increasingly recognized as a prosurvival mechanism in response to EGFR inhibition by TKIs that limits the efficacy of TKIs in vitro and in vivo [9,10,11,12,13,14,15]. Conversely, inhibition of autophagy in EGFR TKI-resistant cell lines sensitizes cells to TKIs and increases cell death [11,14,16].

High glycolytic flux to meet the elevated tumor demands for energetic and biosynthetic precursors needed for survival and proliferation requires the proper function of 6-phosphofructo-2-kinase/fructose-2,6-bisphosphatase 3 (PFKFB3) [16]. The PFKFB family of enzymes are essential glycolytic activators whose function is to catalyze the synthesis and degradation of fructose-2,6-bisphosphate (F26BP)—an allosteric activator of 6-phosphofructo-1-kinase (PFK-1), a rate-limiting enzyme and essential control point in the glycolytic pathway [17,18]. PFKFB3 is overexpressed in human lung, breast, prostate tumors and is selectively required for the survival and growth of transformed cells [19,20]. Moreover, PFKFB3 expression can be regulated by oncogenes like hypoxia-inducible factor 1-alpha (HIF-1a), RAS, mesenchymal epithelial transition (MET), phosphoinositide 3-kinases (PI3K/Akt), and phosphatase and tensin homolog (PTEN) to provide metabolic control of tumor development [21,22,23,24]. It has been shown that genetic or pharmacological inhibition on PFKFB3 attenuates glycolytic flux and induces prosurvival autophagy in colon and gynecological cancer in vitro and in vivo, thus limiting tumor’s sensitivity to chemotherapy [25,26,27,28,29,30]. Recently it has been shown that the small-molecule inhibitor of PFKFB3, PFK158, reduces F26BP production, glycolytic flux, glucose uptake, ATP production, and lactate release, and suppresses cell proliferation in different types of cancer cells by inducing the autophagic flux in vitro and in vivo [27,31,32]. PFKFB3 expression and spatial localization contributes to autophagy flux and its effect depends on the type of stimulus that induced autophagy [31]. Moreover, PFKFB3 expression is inversely related to autophagy levels [32]. However, the role of PFKFB3 in regulating autophagy-dependent cell survival in lung cancer cells has not been established.

We previously demonstrated that dual inhibition of PFKFB3 and EGFR with small-molecule inhibitors synergistically improves erlotinib cytotoxicity in NSCLCs with different EGFR mutation status in vitro [33]. Moreover, we showed that PFKFB3 is selectively required to maintain cell survival in response to EGFR inhibition in NSCLCs. Given that PFKFB3 and EGFR are potent regulators of basal and drug-induced autophagy, we sought to evaluate the possible crosstalk between PFKFB3 and EGFR in regulating erlotinib-induced autophagy in NSCLCs. Here, we show that the addition of the PFKFB3 inhibitor PFK158 abrogates the cytoprotective autophagy flux induced by erlotinib resulting in the accumulation of autophagic vacuoles in the cytosol in NSCLC. Our studies support a model in which EGFR inhibition promotes a high autophagic flux, which is disrupted by PFKFB3 inhibition. Thus, our studies reveal a novel link between PFKFB3, EGFR and autophagy that supports a drug-resistant state in NSCLCs.

## 2. Materials and Methods

### 2.1. Reagents

Chloroquine (C6628) was purchased from Sigma-Aldrich (St. Louis, MI, USA). Erlotinib (S1023, S7786) was obtained from Selleckchem. PFK158 (HY-12203) was purchased from MedChem Express.

### 2.2. Cell Culture

H522 and PC9 cells were purchased from the American Type Culture Collection (ATCC). Cells were cultured in RPMI (Invitrogen, Waltham, MA, USA) supplemented with 10% fetal bovine serum (FBS, Clontech, Mountain View, CA, USA) and 50 μg/mL gentamicin (Life Technologies, Carlsbad, CA, USA). Cells were incubated at 37 °C with 5% CO_2_.

### 2.3. Antibodies and Western Blotting

Whole cell lysates were harvested using RIPA buffer (ThermoFisher, Waltham, MA, USA) supplemented with protease inhibitors. Protein concentration was determined using the BCA protein assay kit (ThermoFisher) following the manufacturer’s instructions. Proteins were separated on 10% or 4–20% Mini PROTEAN or CRITERION TGX gels under reducing conditions and transferred to Immun-Blot PVDF membranes (Bio-Rad, Hercules, CA, USA). The membranes were blocked with 5% BSA or 5% nonfat milk in TBS-T (0.1% Tween20) and immunoblotted with the indicated antibodies. HRP-conjugated goat anti-rabbit (Invitrogen) and anti-mouse IgG (Sigma) were used as secondary antibodies. Amersham ECL Prime Western blotting detection reagent (GE Healthcare, Chicago, IL, USA) was used to detect immunoreactive bands. The membranes were visualized on autoradiography film BX (MidSci, Valley Park, MO, USA). Antibodies to detect p62 (8025) and phospho-AMPK (2535) were purchased from Cell Signaling Technology. LC3 B I/II (L7543) and β-actin antibodies were purchased from Sigma-Aldrich. Total AMPK antibody (3759) was purchased from Abcam. Quantitative densitometry was performed with ImageJ (NIH, Bethesda, MD, USA), and signal density was normalized to the corresponding β-actin loading control.

### 2.4. Growth Rate Assay

Cells were seeded in triplicates at a density of 1 × 10^4^ cells/well in 96-well plates 24 h prior to the addition of treatment. Cell numbers were calculated using the FluoReporter dsDNA quantitation kit (Molecular Probes, Eugene, OR, USA) at the time of treatment (0 h) and 24 h after treatment following the manufacturer’s instructions. To evaluate the drug effect on cell survival, GR values were calculated using the equation: GR(*d*) = 2∧(log2(*x*_d_/*x*_0_)/log2(*x*_ctrl_/*x*_0_)) − 1, where *x*_d_ and *x*_ctrl_ = amounts of cells after treatment with drug (d) or vehicle (ctrl); *x*_0_ = cell amount at the time of treatment [34]. The difference in cell number in the presence of chloroquine (CQ) was calculated using the equation: Number of cells = N_d+CQ_ − N_d-CQ_, where N_d+CQ_ = amount of cells after treatment (erlotinib or/and PFK158) in combination with CQ; N_d-CQ_ = amount of cells after treatment (erlotinib or/and PFK158) in the absence of CQ. Data are presented as mean ± SD of two independent experiments with technical triplicates.

### 2.5. GFP-LC3 Visualization

Plasmid vector containing green fluorescent protein linked to microtubule-associated protein 1 LC3 was a generous gift from Dr. Gomez-Gutierrez. PC9 and H522 cells were seeded in 6-well plates at a density of 1 × 10^5^ cells/well and transfected with GFP-LC3 using JetPRIME transfection reagent (Polyplus transfection) 24 h before treatment. The expression of GFP was monitored by fluorescence microscopy 24 h (H522) or 36 h (PC9) after treatment. Images were taken at 20× magnification with the EVOS FL Imaging System (Thermo Fisher Scientific). Cells were classified as having a predominantly diffuse GFP stain or having numerous punctate structures representing autophagosomes [35]. GFP-LC3 puncta were quantified by counting the number of GFP-positive puncta per cell in random cells per multiple random microscopic fields using ImageJ (NIH) (90 cells/treatment regimen/experiment). Data are presented as mean ± SD of two independent experiments.

### 2.6. Acridine Orange Staining

Acridine orange (AO) staining was employed to visualize acidic vesicular organelles. AO (Sigma-Aldrich) was dissolved in PBS (5 μg/mL). H522 and PC9 cells were seeded in 6 plates at a density 1 × 10^5^ cells/well and treated with appropriate inhibitors for 48 h and then incubated with AO at 37 °C for 10 min. After 2 washes with PBS, images were taken immediately at 20× magnification with the EVOS FL Imaging System (Thermo Fisher Scientific) [36]. Vesicle amount per cell was quantified by counting the number of AO vesicles per cell in bright field images in random cells per multiple random microscopic fields using ImageJ (NIH) (90 cells/treatment regimen/experiment). Data are presented as mean±SD of two independent experiments.

### 2.7. Statistics

Results are reported as the mean ± SD. *p*-values were calculated using two-way ANOVA with Tukey’s post hoc analysis using Graphpad Prism, version 9.0.1 (GraphPad, San Diego, CA, USA). *p*-values < 0.05 were considered to be statistically significant.

## 3. Results

### 3.1. PFKFB3 Inhibition Limits the Cytotoxic Effects of Chloroquine in Erlotinib-Treated mutEGFR NSCLCs

NSCLCs activate autophagy in response to erlotinib thus limiting its cytotoxicity in these cells [12,37]. Our previous report demonstrated that the addition of a small molecule antagonist of PFKFB3, termed PFK158, to erlotinib treatment significantly decreased the survival of NSCLC cells [33]. Here, we sought to evaluate the contribution of PFKFB3 and autophagy inhibition on cell proliferation/viability in NSCLC cells treated with erlotinib. We utilized NSCLC PC9 cells, which harbor an activating mutation of EGFR (mutEGFR) that renders them highly sensitive to the cytotoxic effects of EGFR TKIs such as erlotinib, and wild-type (WT) EGFR H522 cells that are modestly sensitive to TKIs. First, we exposed either mutEGFR (PC9) or WT EGFR (H522) to erlotinib in the presence or absence of the PFKFB3 inhibitor PFK158 for 24 h and measured cell proliferation. In line with EGFR mutation status, we found that erlotinib treatment (0.5 μM) in mutEGFR PC9 cells caused a significant growth inhibition effect, which was further increased by the addition of PFK158 (Figure 1A, left panel). In contrast, WT-EGFR H522 cells were relatively insensitive to the effects of erlotinib (2 μM) although we did observe growth inhibition upon the addition of PFK158 (Figure 1A, right panel). Next, we assessed the contribution of autophagy to NSCLC survival by exposing the cells to chloroquine (CQ)—an agent that inhibits autophagy flux by suppressing lysosomal acidification and preventing the degradation of autolysosome content [38]. We treated NSCLC cells with chloroquine, either alone or in combination with erlotinib, PFK158 or erlotinib plus PFK158 and compared cell numbers 24 h post treatment. Exposure of PC9 cells to chloroquine decreased the viability of the cells exposed to erlotinib or PFK158 but failed to decrease cell viability in the combination treatment, suggesting that PFKFB3 inhibition impairs cytoprotective erlotinib-induced autophagy (Figure 1B, left panel). We also found that chloroquine significantly increased cytotoxicity across all treatments in WT-H522 cells, indicating that basal autophagy is essential for maintaining cell viability in EGFR WT cells (Figure 1A, right panel). Autophagy is a dynamic process that starts with the formation of autophagosomes followed by the fusion of the autophagosome and lysosome and ends with the degradation of the autophagosome contents by lysosomal hydrolases. We assessed the levels of the autophagy marker LC3 B II (lipidated form of LC3 B), which is commonly used to monitor the amount of autophagosome formed in the cells, in the presence and absence of chloroquine. Immunoblotting revealed that exposure to erlotinib promoted the conversion of LC3 II in both mutPC9 and WT-EGFR H522 cells, indicating that erlotinib promotes autophagosome formation in these cells (Figure 1C, treatment with no CQ). These results are in agreement with previously published data demonstrating elevated autophagy in response to TKI treatment in NSCLCs [11,12]. In the presence of chloroquine, we observed increased LC3 B II levels due to a decrease in autophagosome turnover. In addition, we observed elevated LC3 II conversion in control treated H522 cells compared to PC9 cells, which was further increased by the addition of chloroquine confirming elevated basal autophagy. Finally, the decreased amount of PC9 and H522 cells when exposed to CQ alone confirmed high dependency of NSCLCs on basal autophagy for survival. Overall, these results indicate that PFKFB3 is required for the ability of the autophagy inhibitor chloroquine to increase sensitivity to erlotinib in NSCLCs.

### 3.2. PFKFB3 Inhibition Promotes Autophagosome Formation in Erlotinib-Treated NSCLCs

Given that chloroquine is ineffective at improving the response of PC9 cells to the combination treatment of erlotinib and PFK158, we next aimed to determine the effects of PFKFB3 inhibition on autophagy initiation. To test this, we assessed the formation of autophagosomes in NSCLCs by monitoring the conversion of cytoplasm-diffuse GFP-LC3-I to punctate forms of membrane-associated GFP-LC3-II, which indicates LC3-II incorporation into the autophagosomes (Figure 2A). While EGFR or PFKFB3 inhibition displayed a moderate increase in GFP-LC3 puncta in both PC9 and H522 cells, the addition of PFK158 to erlotinib led to a marked increase in green puncta per cell (Figure 2A,B) suggesting an accumulation of autophagosomes. Degradation of the autophagosomal content occurs at the late stages of autophagy; thus, measurement of cargo turnover is used to monitor the completion of the autophagy process. Cargo carrier protein p62 links ubiquitinated proteins to LC3 in the formed autophagosome and is targeted for degradation along with the autophagosome content in the autolysosome at the late stages of autophagy flux [40]. Accordingly, p62 accumulates in cells when autophagy flux is inhibited. In order to get more insight on the effect of PFFKB3 inhibition on erlotinib-driven autophagy, we analyzed the levels of the autophagosome substrate p62. In agreement with previous observations, erlotinib treatment caused a significant decrease in p62 expression in both PC9 and H522 cells compared to control indicating that erlotinib triggers autophagy [12]. Notably, we observed a small increase in p62 levels in NSCLC cells treated with erlotinib plus PFK158 compared to erlotinib alone suggesting that the combination may have a suppressive effect on autophagy at the late stages. Taken together, these results suggest that the simultaneous inhibition of EGFR and PFKFB3 may affect the completion of autophagy either by inducing a delay in late autophagosome maturation or by blocking the autolysosome disintegration.

### 3.3. PFKFB3 Inhibition Blocks Erlotinib-Induced Turnover of p62

Increased autophagosome amounts (as shown by elevated LC3 II levels) can indicate either an induction in autophagy or a block at the late stages of the autophagy pathway [40]. Our data suggest that erlotinib-induced autophagy flux is impaired by PFK158 as p62 levels were increased upon dual treatment (Figure 2C). Thus, we next sought to test the effect of PFKFB3 inhibition on erlotinib-activated autophagy flux. Autophagy flux is a measure of autophagic degradation activity and thus, measurement of LC3 II conversion and p62 turnover in the presence or absence of lysosomal blockade with chloroquine (which blocks autophagosome and lysosome fusion and neutralize the lysosomal pH) is needed to determine the rate of transit of autophagosome cargo through lysosomal degradation. We blocked autophagosome maturation by performing short-term chloroquine treatment and measured the expression levels of the autophagy-associated markers p62 and LC3 II. Briefly, cells were treated with either PFK158, erlotinib or the combination for 24 h followed by exposure to chloroquine for 2–6 h and p62 and LC3 II expression levels were analyzed. Similar to the results shown in Figure 2C, combination-treated cells displayed increased levels of p62 compared to erlotinib-treated cells (Figure 3A, CQ 0 h). As expected, blockage of autophagy flux promoted the accumulation of p62 in both PC9 and H522 cells treated with either PFK158 or erlotinib alone. However, the addition of chloroquine had no effect on p62 levels in the combination treatment in these cells (Figure 3B). Coincidentally, exposure to dual treatment moderately increased LC3 II accumulation compared to erlotinib alone in PC9 cells after 6 h of exposure to CQ (Figure 3C, left panel). Of note, chloroquine exposure in H522 resulted in a dramatic accumulation of LC3 II and decreased p62 turnover in control-treated cells, confirming higher basal autophagy levels in these cells. Importantly, combinational treatment in H522 cells significantly delayed LC3 and p62 clearance when compared to any individual treatment (Figure 3C, right panel). Given that CQ blocks the fusion of autophagosomes and lysosomes, and that erlotinib and PFK158 co-treatment did not accumulate p62 upon chloroquine treatment, these results suggest that PFKFB3 inhibition impairs erlotinib-induced autophagy at the late stages in NSCLC.

### 3.4. PFKFB3 Inhibition Promotes Accumulation of Intact Acidic Vesicular Organelles in Erlotinib-Treated NSCLC Cells

The late steps of autophagy flux include the fusion of the autophagosome with the lysosome to form the autolysosome, resulting in the degradation of the autophagosomes content and the release of nutrients and metabolites to the cytosol [41]. Given that the addition of PFK158 to erlotinib therapy resulted in the accumulation of autophagosomes, we wanted to further confirm whether PFK158 blocked erlotinib-driven autophagy at late stages in NSCLC cells. We exposed PC9 and H522 cells to individual or combined treatment for 48 h and monitored acidic vesicle accumulation (20 µM CQ was used as a positive control). We monitored late autophagosome maturation by visualizing acidic vesicular organelles (AVOs) using the lysosomotropic dye acridine orange (AO) in live cells. As shown in Figure 4, exposure of PC9 cells to PFK158 or erlotinib markedly increased the number of acidic vesicles (red staining), which was coincident with an increase in yellow staining corresponding to the areas with protonated AO released from lysosomes (Figure 4A) [42]. However, we observed a dramatic increase in red puncta in dual treated PC9 cells suggesting the accumulation of intact autolysosomes (AVOs with completely intact membrane accumulate higher amounts of AO due to lower pH of its content, Figure 4A,B) [36]. Similarly, AVO accumulation and protonated AO release to the cytosol was abrogated upon dual treatment in H522 cells (Figure 5A,B). These results indicate that PFKFB3 inhibition with PFK158 decreased TKI-induced autophagy flux by regulating the maturation of acidic vesicles.

### 3.5. PFKFB3 Inhibition Abrogates Stress-Induced AMPK Activation in Erlotinib-Treated NSCLC Cells

PFKFB3 is in a tight functional cross-talk with AMPK—the main sensor of cellular energy and a major driver of autophagy. AMPK directly phosphorylates PFKFB3 to promote its kinase activity while PFKFB3 can modulate autophagy via regulation of AMPK signaling [31,43]. Recently, it has been shown that AMPK is required for basal lysosome function, including its hydrolytic activity [44]. Moreover, it has been shown that AMPK uses the lysosome as a scaffold for the direct regulation of mTOR1 and that proper AMPK activation requires lysosomal damage [45,46]. Given that AMPK regulates lysosomal function [47] and the significant accumulation of acidic vesicles observed in response to dual treatment in PC9 and H522 cells, we investigated AMPK expression and activation in response to erlotinib or/and PFK158 in both PC9 and H522 cells. We found that both EGFR and PFKFB3 inhibition promoted AMPK activation (indicated by the increase in AMPK phosphorylation at T172) in a dose-dependent manner in both PC9 and H522 cells (Figure 6). In contrast, dual PFK158 and erlotinib treatment abrogated AMPK activation in both cell lines. Considering the key role of AMPK in regulating autophagy and lysosomal function, our data suggests that loss of AMPK activation may contribute to the decrease in erlotinib-driven autophagy caused by PFKFB3 inhibition.

## 4. Discussion

In this study, we demonstrate that PFKFB3 inhibition impairs erlotinib-induced autophagy in NSCLCs with different EGFR mutation status. Adding to the canonical understanding that PFKFB3 expression is essential to maintain elevated glycolytic flux in lung cancer cells to promote proliferation and survival, we provide evidence that PFKFB3 is required for erlotinib-induced cytoprotective autophagy. While the mechanism by which PFKFB3 inhibition blocks erlotinib-induced autophagy remains unknown, the data presented here provides evidence that PFK158 impairs erlotinib-induced autophagy in NSCLCs that in parallel with the other inhibitory effects of PFK158 improves the cytotoxicity of erlotinib in NSCLCs.

Lung cancers with activating mutations of EGFR are treated with TKIs, which target the activity of the receptor. However, moderate response and development of acquired resistance upon EGFR TKI therapy limits the clinical benefit and contributes to the incredibly low survival rates among lung cancer patients. Current efforts to improve treatment in lung cancer patients focus largely on targeting secondary and tertiary mutations in EGFR kinase domain. However, mounting evidence indicates that autophagy can support tumor growth and promote resistance to a variety of therapies, including radiotherapy, cytotoxic and targeted therapies [48,49,50]. Thus, understanding of molecular mechanisms contributing to basal and drug-induced autophagy is paramount for uncovering druggable alterations that may be able to prevent or alleviate resistance to EGFR-TKI in lung cancer patients. We observed that EGFR inhibition with erlotinib induced autophagy in both WT and mutEGFR NSCLC cells. These observations contribute to the already available evidence showing elevated autophagy in response to EGFR inhibition in different types of cancer in vitro and in vivo [10,11,12,51,52,53].

NSCLCs require elevated glucose metabolism to ensure cell survival in response to TKI treatment [54,55,56]. Recently it has been shown that EGFR interacts with and stabilizes the sodium/glucose cotransporter 1 (SGLT1) to maintain intracellular glucose levels and to protect cells from autophagic cell death induced by glucose starvation [57]. PFKFB3 expression drives glycolytic flux while PFKFB3 inhibition promotes the accumulation of reactive oxygen species and induces autophagy [25]. We recently demonstrated that PFKFB3 expression is required for EGFR-mediated glycolytic flux in different NSCLCs and that pharmacological inhibition or PFKFB3 mRNA ablation completely block EGFR-driven glycolysis and significantly attenuate the viability of erlotinib-treated cells [33]. In this study, we discovered that PFKFB3 inhibition in both PC9 and H522 cells exposed to erlotinib alters p62 turnover and leads to a dramatic accumulation of intact acidic vesicular organelles in erlotinib-treated PC9 and H522 cells. It is important to note that while PFKFB3 blocked the completion of erlotinib-driven autophagy, PFKFB3 inhibition alone led to an increase in autophagy revealing a multifaceted and a more complex role of PFKFB3 in the regulation of autophagy than initially described. While the mechanism by which PFKFB3 inhibition promotes autophagy but blocks erlotinib-induced autophagy is not clear, our findings that PFKFB3 inhibition attenuates stress-induced autophagy are in agreement with data published by other groups [16,29,33]. Specifically, it has been shown that PFKFB3 inhibition blocks stress-induced autophagy flux via regulation of AMPK signaling [31] or via direct interaction with members of the autophagy machinery, including p62/sequestostome 1 [29,32]. In addition, Almacellas et al. showed that PFKFB3 regulates the translocation of mTORC1 to lysosomes by direct interaction with Rag B suggesting the potential regulation of lysosomal function by PFKFB3 [58]. Moreover, it has been shown that both PFKFB3 ablation and inhibition with PFK158 trigger late endosome formation in malignant pleural mesothelioma by increasing the interaction of Rac1 with Rab7 [59], where Rab7 plays an important role in the final maturation of late autophagic vacuoles [60]. Although, additional studies are required to elucidate the molecular mechanism underlying the impaired autophagosome maturation and to track the autolysosome release to the cytosol in real time, the data shown here demonstrate for the first time that PFKFB3 inhibition impairs erlotinib-induced autophagy flux by compromising the late steps of autophagy.

Under metabolic or energy stress conditions, the LKB1-AMPK axis inhibits the anabolic pathways and activates the catabolic pathways to maintain metabolic homeostasis for cell survival. In our studies, we observed elevated AMPK activity in response to EGFR or PFKFB3 inhibition. These findings are in agreement with prior reports demonstrating AMPK activation in response to EGFR or PFKFB3 inhibition [25,61,62]. However, dual PFK158 and erlotinib treatment abrogated AMPK activation in both PC9 and H522 cells. We speculate that prolonged cell stress upon dual PFKFB3 and EGFR inhibition leads to deregulated AMPK function which, in turn, might contribute to the impaired autophagy flux observed in the combo-treated cells. Recently, it has been shown that prolonged glucose deprivation results in AMPK deregulation that interrupts metabolic adaptation and causes rapid cell death [63]. Moreover, recent studies have shown that in LKB1-mutant cells, glucose starvation elicits an oxidative stress, which causes AMPK protein oxidation and inactivation, and eventually cell death [64]. The data shown suggest that PFKFB3 inhibition modulates the activity of AMPK in erlotinib-treated cells. These results are in line with a previous report showing that inhibition of PFKFB3 decreased AMPK phosphorylation and autophagy in response to H_2_O_2_ stimulation in a renal carcinoma cell model [31]. Our studies warrant further investigation to elucidate the precise mechanism by which combined blockade of PFKFB3 and EGFR results in decreased AMPK activation.

Importantly, the inhibition of autophagy flux with chloroquine confirmed the dependency of NSCLCs on basal autophagy for survival (Figure 1) [65,66]. Given that we previously observed a significant improvement in TKI cytotoxicity in WT EGFR cells upon PFKFB3 inhibition in vitro, we speculate that basal autophagy in NSCLCs may require PFKFB3 expression. Further studies are required to delineate the precise mechanisms by which PFKFB3 supports basal autophagy. Our observations contribute to the increasing evidence suggesting the exclusive importance of autophagy in tumorigenesis [67,68,69].

A limitation of our studies is the use of only one mutEGFR and one WT-EGFR NSCLC cell line and thus it is possible that our findings are cell line specific. Further investigation of NSCLCs with different EGFR mutation statuses will be needed to determine whether the role of PFKFB3 in regulation of cytoprotective autophagy is context dependent.

These findings provide the rationale for future studies testing the utility of targeting PFKFB3 to improve cytotoxicity in NSCLCs with intrinsic resistance to chemotherapy.

In summary, our studies are the first to identify PFKFB3 as a molecular regulator of the erlotinib-induced autophagy flux in NSCLCs. Our findings imply that exploiting PFKFB3 as an autophagy flux mediator for therapeutic purposes could represent a novel combination treatment regimen against lung cancer tumors with intrinsic or acquired resistance to EGFR TKIs.

## Figures and Tables

**Figure 1 cells-10-01679-f001:**
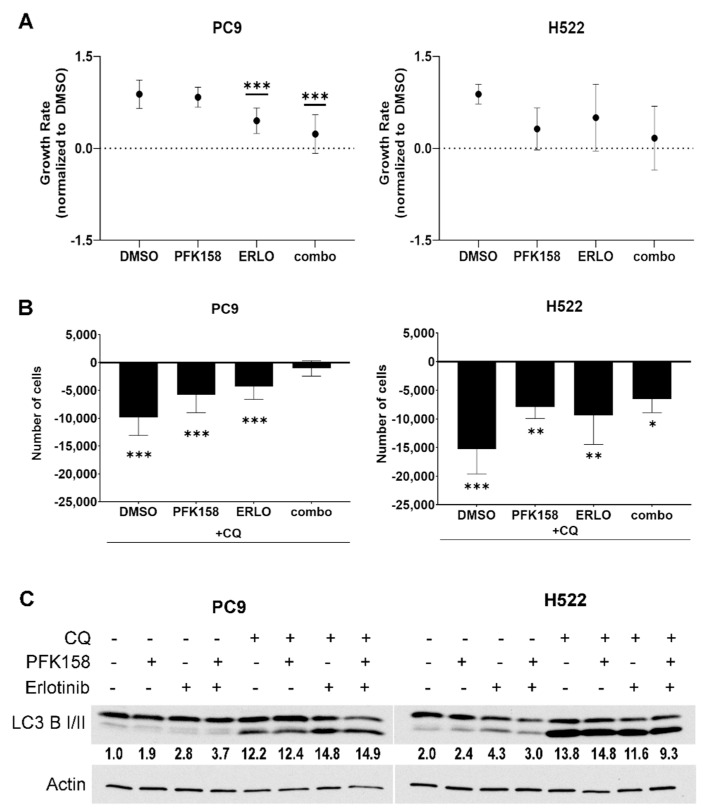
Basal and drug-induced autophagy is essential for cell growth in NSCLCs exposed to EGFR or PFKFB3 inhibitors. PC9 and H522 cells were exposed to erlotinib (0.5 μM or 2 μM, respectively) and/or PFK158 (5 μM or 1 μM, respectively) alone or in combination with the autophagy inhibitor chloroquine (CQ, 100 µM) for 24 h. Growth rates (GR) were calculated as indicated in Methods, where GR > 0 indicates inhibited growth, GR = 0, cytostatic effect and GR < 0, cytotoxic effect. Error bars, mean ± S.D. of two independent experiments (n = 6). *p* values are shown as follows: *** < 0.001 (compared to DMSO treatment) (**A**). The difference in number of cells in response to the presence of chloroquine in appropriate treatment was calculated and presented as delta in number of cells. Error bars, mean ± S.D. of two independent experiments (n = 6). *p* values are shown as follows: * < 0.05; ** < 0.01; and *** < 0.001 (compared to appropriate treatment without CQ) (**B**). PC9 and H522 cells were exposed to 100 µM CQ for 4 h and levels of LC3 B II were evaluated in immunoblotting and are representative from three independent experiments (**C**). LC3 B II levels presented as normalized to actin and to the control treated cells (DMSO, PC9) [39].

**Figure 2 cells-10-01679-f002:**
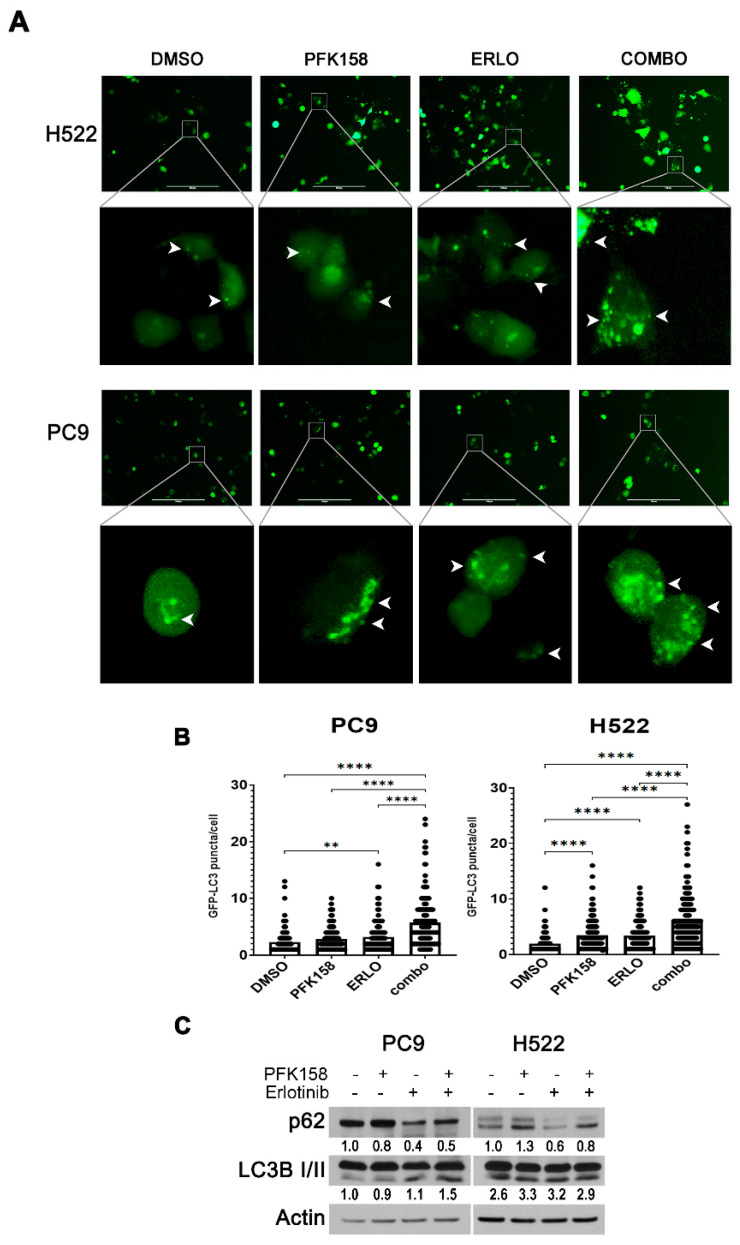
LC3-GFP overexpression reveals accumulation of autophagosomes in response to dual treatment in NSCLCs. PC9 and H522 cells were transfected with a pEGFP-LC3 plasmid and treated with erlotinib (PC9: 0.5 µM; H522: 2 µM) and/or PFK158 (PC9: 5 µM; H522: 0.75 µM) for 36 h or 24 h, respectively. (**A**) GFP-LC3 localization was evaluated by fluorescence microscopy. The formation of autophagosomes is depicted by the formation of puncta structures. Images were taken at 20× magnification with an EVOS FL Imaging System microscope. Scale bar: 100 µm. (**B**) Quantification of GFP-LC3 puncta/cell. The data presented are compiled counts of 180 cells/treatment regimen. Error bars, ±SD of two independent experiments. *p* values are shown as follows: ** < 0.01, **** < 0.0001 (compared to indicated treatment). (**C**) Whole cell lysates were immunoblotted with the indicated antibodies. Images shown are representative of two independent experiments. LC3 B II levels presented as normalized to actin and to the control treated cells (DMSO, PC9).

**Figure 3 cells-10-01679-f003:**
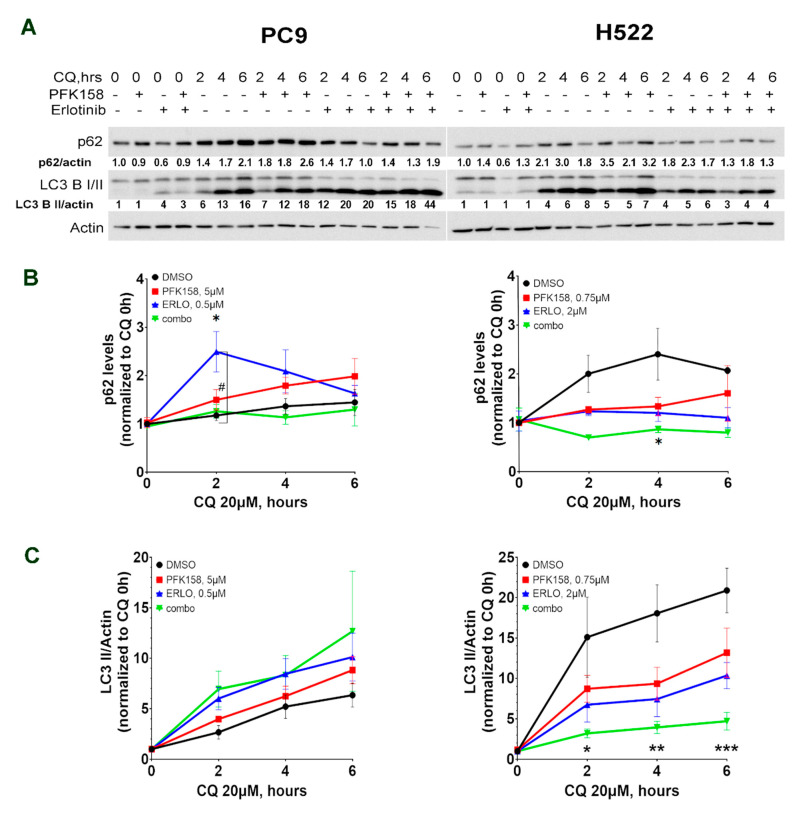
PFKFB3 inhibition impairs p62 turnover in TKI-treated cells in NSCLCs. PC9 and H522 cells were treated with erlotinib (PC9: 0.5 µM; H522: 2 µM) and/or PFK158 (PC9: 5 µM; H522: 0.75 µM) alone or in combination for 24 h followed by exposure to 100 µM CQ for 0–6 h. Whole cell lysates were immunoblotted with the indicated antibodies. Images shown are representative of three independent experiments. LC3 B II and p62 levels presented as normalized to actin and to the control treated cells (DMSO, PC9 or H522) (**A**). p62 levels were normalized to loading control (actin) and to the levels of p62 for each treatment regimen before to exposure to CQ (CQ 0 h) (**B**). The LC3 II/I ratio was normalized to the LC3 II/I ratio in the appropriate treatment before to exposure to CQ (CQ 0 h) (**C**). Error bars, ±SD of three independent experiments (three-way ANOVA with Tukey’s multiple comparisons test). *p* values are shown as follows: * < 0.05, ** < 0.01, *** < 0.001 (compared to DMSO treatment or as indicated); # < 0.05 (combo compared to erlotinib treatment).

**Figure 4 cells-10-01679-f004:**
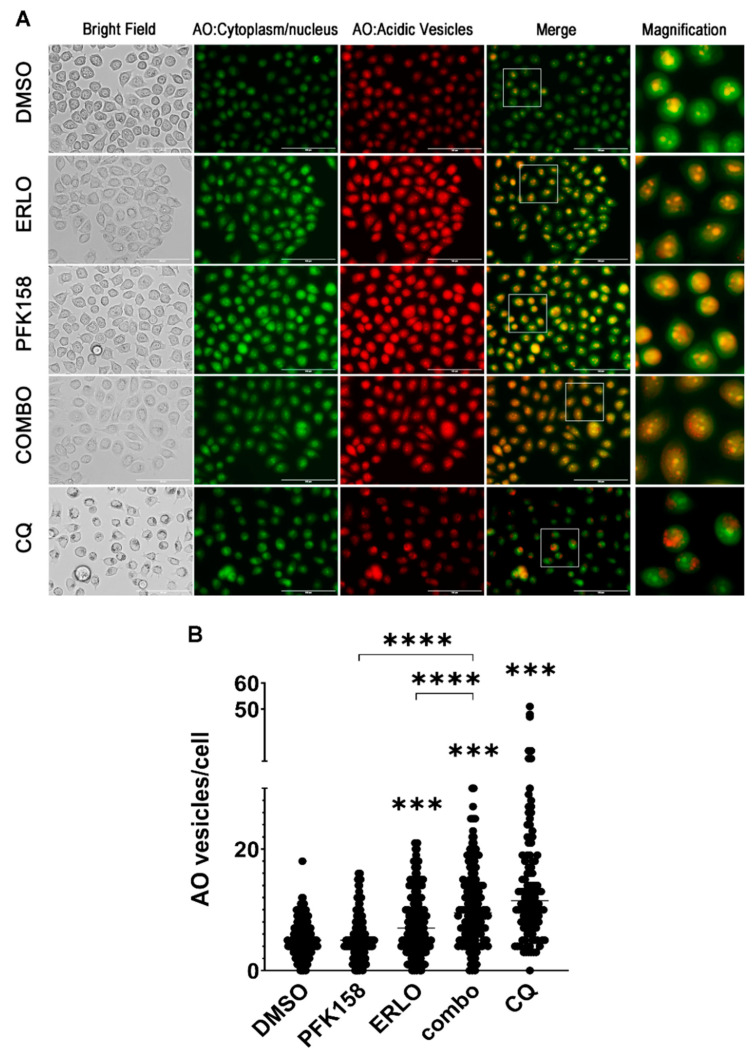
Dual treatment promotes the accumulation of non-released acidic vesicular organelles (AVOs) in mutEGFR PC9 cells. PC9 cells were exposed to 0.5 µM Erlotinib and 5 µM PFK158 alone or in combination for 48 h. Cells treated with chloroquine (CQ, 20 µM) were used as positive control for proper AVO accumulation and labeling. AVO accumulation was evaluated by acridine orange staining. Images were taken at 20× magnification with an EVOS FL Imaging System microscope (**A**). Images are representative of three independent experiments. Bright field images illustrate cell morphology and AO accumulation in dark vesicles. Under fluorescence, acridine orange staining in the cytoplasm and nucleus fluoresce green, whereas the acidic compartments fluoresce bright red or orange-red. Scale bar: 100 µm. The number of AO positive vesicles per cell was counted using Image J (**B**), *p* values are shown as follows: *** < 0.001, **** < 0.0001 (compared to DMSO or indicated treatment).

**Figure 5 cells-10-01679-f005:**
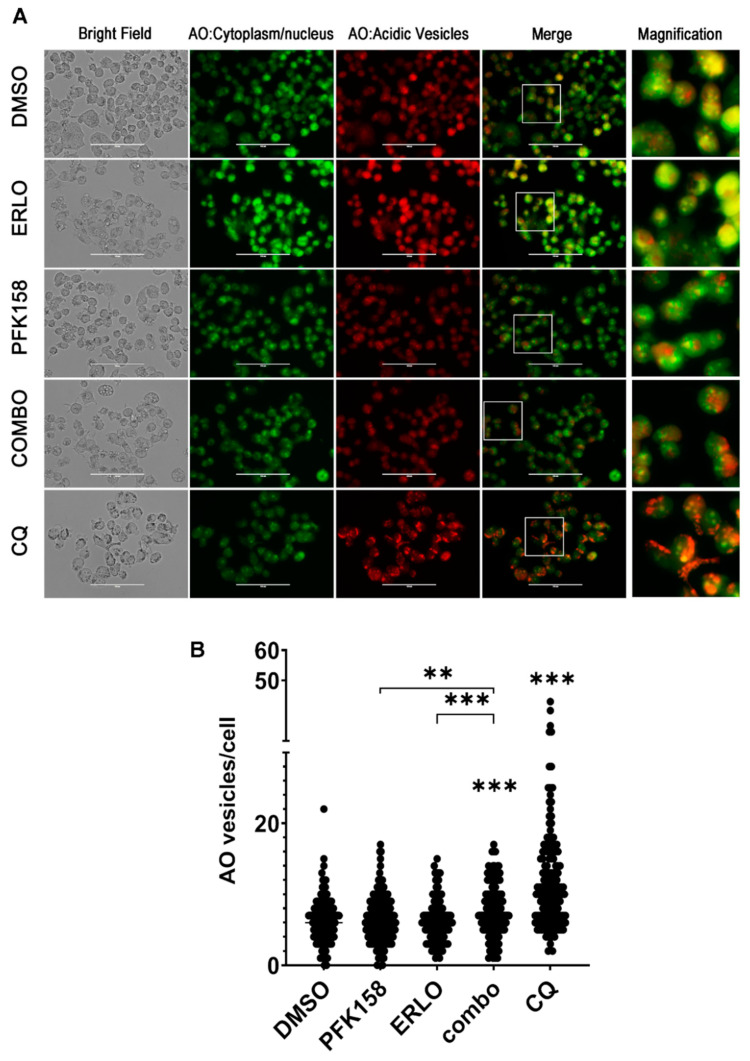
Dual treatment promotes the accumulation of non-released acidic vesicular organelles (AVOs) in WT EGFR H522 cells. H522 cells were exposed to 2 µM erlotinib and 0.75 µM PFK158 alone or in combination for 48 h. Cells treated with chloroquine (CQ, 20 µM) were used as positive control for proper AVO accumulation and labeling. AVO accumulation was evaluated by acridine orange staining. Images were taken at 20× magnification with an EVOS FL Imaging System microscope (**A**). Images are representative of two independent experiments. Bright field images illustrate cell morphology and AO accumulation in dark vesicles. Under fluorescence, acridine orange staining in the cytoplasm and nucleus fluoresce green, whereas the acidic compartments fluoresce bright red or orange-red. Scale bar: 100 µm. The number of AO positive vesicle per cell was counted using Image J (**B**), *p* values are shown as follows: ** < 0.01, *** < 0.001, (compared to DMSO or indicated treatment).

**Figure 6 cells-10-01679-f006:**
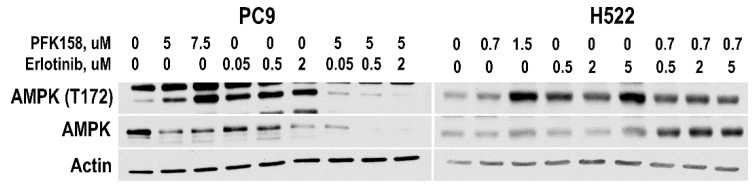
AMPK expression and phosphorylation in NSCLCs exposed to EGFR or PFKFB3 inhibitors. PC9 and H522 cells were exposed to different concentrations of erlotinib and PFK158 alone or in combination for 48 h. Levels of AMPK were evaluated by immunoblotting.

## Data Availability

Not applicable.

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
