# Peer review of "PFKFB3 Inhibition Impairs Erlotinib-Induced Autophagy in NSCLCs"

_cells, 2021, doi:10.3390/cells10071679_

Round 1

Reviewer 1 Report

The current article entitled," PFKFB3 Inhibition Impairs Erlotinib-Induced Autophagy in 2 NSCLCs is interesting and has scientific merits may be considered for publication.

Here authors reported the effect of   pharmacological inhibition of PFKFB3 on erlotinib-driven autophagy in NSCLC cells with different mutation status. They identified that PFKFB3 as a mediator of erlotinib-induced autophagy in NSCLCs. Furthermore, they shows that PFKFB3 inhibition sensitizes NCSLCs to erlotinib via impairing autophagy flux. In summary, this interesting research uncovered a novel crosstalk between PFKFB3 and EGFR that regulates erlotinib induced autophagy thus contributing to erlotinib sensitivity in NSCLCs. Altogether paper is well written.

Which methods was used to analyze viability of the cells exposed to erlotinib or PFK158? Use more conceiving methods such as MTT or XTT assay.

When cells exposed to erlotinib or PFK158 is there any evidence of apoptosis? 

NSCLCs require elevated glucose metabolism to ensure cell survival does erlotinib or PFK158 have any effect on glycolysis of NSCLCs cells? OCR and ECAR?

PFK158 may be modulating AMPK signaling? Which in turn decreasing autophagy flux in NSCLCs cells. 

PFKFB3 inhibition promotes autophagosome formation in erlotinib-treated NSCLCs.

To determine autophagy flux, they should use mRFP (mCherry)- GFP tandem fluorescent-tagged LC3 and determine red-green ratio by using flow cytometry. Best possible way is to use CAG-RFP-EGF-LC3 reporter mice to determine autophagy flux.

Reagent, antibody catalog number is missing in the methods section. 

Scale bar are missing in figure 2A

Reviewer 2 Report

Using 2 NSCLC cell lines with a different EGFR status, the authors showed that PFKFB3 is required for erlotinib-induced autophagy in NSCLC independently of their EGFR status and that PFKFB3 inhibition sensitizes NSCLC cells to erlotinib via late stage autophagy inhibition. While the overall study is well executed, there are some limitations and questions: 

Major comments:

  • In order to make sure that the effects observed are not cell line dependent, the authors should repeat some of the main experiments on another Mut- and WT-EGFR cell lines.
  • Fig 1A:
    • I am not sure this is the best way to present the data. Would it be possible to also present the proliferation/viability for each condition, in addition to the delta graph?
    • It looks like CQ alone affect the proliferation better than its combination with PFK158 or Erlotinib. The authors do not mention it in the manuscript, it would be good to add a sentence about it.
    • The authors state that “Exposure of PC9 cells to chloroquine decreased the viability of the cells exposed to erlotinib or PFK158 but failed to decrease cell viability in the combination treatment, suggesting that PFKFB3 inhibition impairs cytoprotective autophagy”. If PFKFB3 inhibition really impaired cytoprotective autophagy, I would not expect CQ to affect proliferation/viability during PFK158 treatment. How could you explain it?
  • Fig 1B: The authors should better describe the western blotting to highlight the effect of each treatment on the autophagic flux. Authors stated that ‘exposure to erlotinib promoted the conversion of LC3 II in both mutPC9 and WT-EGFR H522 cells indicating that erlotinib promotes autophagosomes formation in these cells (Figure 1B)’. I do not agree when you compare the CQ samples, it looks like Erlotinib only induces autophagy in PC9 cells (12.2 vs 14.8) and not in H522 cells (13.8 vs 11.6).
  • Fig 2A-B: Line 187-188, the authors stated that “the addition of PFK158 to erlotinib led to a marked increase in green puncta per cell (Figure 2 A,B) indicating an accumulation of autophagosomes.” I would use the word “suggesting” instead of “indicating” as you cannot attest that unless you are using CQ.
  • Fig 3: The way of presenting the p62 and LC3 quantification (B and C) is somewhat confusing and difficult to understand. Maybe it would be better to present it the same way as the other western blotting (Fig 2C for example) with numbers under the blot.
  • Line 237-240: The authors said “Given that CQ blocks the fusion of autophagosomes and lysosomes, and that erlotinib and PFK158 co-treatment did not accumulate p62 upon chloroquine treatment, these results suggest that PFKFB3 inhibition impairs erlotinib-induced autophagy at the late stages in NSCLC.” I disagree with this conclusion, it is too early at that point to say that as an inhibition in the early stage of autophagy would also lead to an accumulation of P62. The authors should modify their conclusion in order to highlight that.
  • Line 258-259: The authors said “whether PFK158 blocked the erlotinib-driven autophagic flux in NSCLC cells”. I would say late stage autophagy or autophagolysosome maturation instead of autophagic flux.
  • Fig 5-6: While the acridine orange assay are pretty convincing, it would be nice to perform supplemental experiments to directly study the autophagolysosomes and not the lysosomes by itself using for example the GFP-RFP-LC3 probe or by studying the colocalization between LC3 and LAMP1. This will strengthen the conclusion suggesting that inhibition of PFKFB3 inhibits the late stage of autophagy. 

Minor comments:

  • Line 58: Only the abbreviation is indicated for HIF-1a, PI3K/Akt, PTEN. Please indicate the full name.

Round 2

Reviewer 1 Report

Thank you for addressing all previous comments. 
Still, I feel the authors could have tried to analyze autophagy flux, as suggested. No matter if the results are negative.  I strongly feel that this experiment must be performed by using  CAG-RFP-EGF-LC3 reporter mice or probes to determine autophagy flux.
They observed that over-expression of GFP-LC3 is toxic for PC9. By using CAG-RFP-EGF-LC3 primary cells from mice will not have this over-expressing issue. Before publishing, I would like to see these results.

Author Response

This manuscript is a resubmission of an earlier submission. The following is a list of the peer review reports and author responses from that submission.

Round 1

Reviewer 1 Report

The revised and strongly reworked article of Lypova et al. is clearly improved and more focused.
However, I still see a major problem in the interpretation of the autophagy experiments, which in my view do NOT support the main conclusion of the authors. 

While the authors show that autophagy inhibition by CQ can impair growth of PC9 cells, the data do not support the conclusion that PFK158 acts via this same mechanism of autophagy inhibition. Moreover, the data presented  confirm previous findings that erlotinib induces autophagy with increased autophagic flux in the tested cell lines (new data presented in Figure 3 confirms this), yet the combination with PFK inhibitor PFK158 does NOT change this response, in contrast to what the authors try to claim. In detail: While erlotinib treatment increases LC3-II/Actin levels in both  untreated and CQ treated cells (which indicates increased autophagy flux by erlotinib), PFK158 does NOT change this response. After 2 and 4 hours incubation, CQ still leads to the accumulation of LC3-II in  erotinib+PFK158 cells at a comparable level to the single erlotinib-treated condition. Tehrefore, autophagy flux is not changes by PFK158 in erlotinib-treated cells. Since p62 can be transcriptionally regulated independent of autophagic acitivity, this marker can be misleading and interpretations should be mainly based on LC3-II levels normalized by Actin or other loading controls!

Overall, the authors should consider that PFK158 elicits growth inhibitory effects on PC9 cells similar to and in parallel to autophagy inhibition, but not through autophagy inhibition.

Reviewer 2 Report

The paper looks good for publication

Reviewer 3 Report

Lypova et al. wrote a manuscript about the effect of PFKFB3 inhibition in erlotinib-induced autophagy in NSCLC cell lines. They used the  EGFR mutant cell line PC9 that is sensitive to erlotinib and an EGFR wild type cell line H522 that is modest senstitive to erlotinib.  The authors used various methods including western blotting and immunofluorescence to demonstrate autophagy induction by erlotinib which was attenuated by PFKFB3 inhibition with  PFK158.   

The results with cloroquine as an inhibitor of autophagic flux are clear but all drug related effects at the level of autophagy are not convincing in my opinion. The western blots hardly show LC3BII accumulation, the puncta structures visualized with EGFP-LC3 are not very clear and finally the acridine orange staining seems to stain predominantly  the nuclei and nucleoli.

Major comments:

  1. Why was a short incubation with erlotinib (24hrs) used and not longer to increase differences between control and erlotinib and between the two cell lines? The authors only used two biological replicates performed in triplicate, stll it was giving a very large SD? Why were different concentrations of the PFKFB3 inihibitor PFK158 used for the two cell lines? Did the authors perform a survival assay with different concentrations of  PFK158 and did it effectively inhibited PFKFB3?
  2. Figure 1B is not easy to understand. What is displayed. Why not fusing figure A and B? Are the H522 faster or slower growing cells?
  3. In figure 2 the number of puncta per cell seem to be similar between control and PFK158 and erlotinib. Means are not visible and the number of counted cells is not indicated. Moreover the puncta are almost not visible with the monotreatments. What was the effect of CQ in this assay?  
  4. Why is LC3BII going down in H522 after erlotinib or combi treatment + CQ (Figure 3)?
  5. What is counted in Figure  4 and 5? Nuclei and nucleoli are also in it? Based on which data/counting  are the statistics performed, since it suggests to me no differences, especially in H522 cells?
  1. The authors use different times of treatment and different concentrations of CQ, why?

Minor comment:

In legend of Figure 1 A-C are not indicated at the beginning of a sentence.